# Large-Scale Piezoelectric-Based Systems for More Electric Aircraft Applications

**DOI:** 10.3390/mi12020140

**Published:** 2021-01-28

**Authors:** Tran Vy Khanh Vo, Tomasz Marek Lubecki, Wai Tuck Chow, Amit Gupta, King Ho Holden Li

**Affiliations:** 1Rolls-Royce@NTU Corporate Lab, Nanyang Technological University, Singapore 637460, Singapore; votran.vk@ntu.edu.sg (T.V.K.V.); rr-tomasz@ntu.edu.sg (T.M.L.); A.Gupta@ntu.edu.sg (A.G.); 2School of Mechanical and Aerospace Engineering, Nanyang Technological University, Singapore 639798, Singapore; wtchow@ntu.edu.sg

**Keywords:** piezoelectric stack, amplification mechanism, quasi-static stepped system, ultrasonic system, piezoelectric-hydraulic, aerospace applications

## Abstract

A new approach in the development of aircraft and aerospace industry is geared toward increasing use of electric systems. An electromechanical (EM) piezoelectric-based system is one of the potential technologies that can produce a compactable system with a fast response and a high power density. However, piezoelectric materials generate a small strain, of around 0.1–0.2% of the original actuator length, limiting their potential in large-scale applications. This paper reviews the potential amplification mechanisms for piezoelectric-based systems targeting aerospace applications. The concepts, structural designs, and operation conditions of each method are summarized and compared. This review aims to provide a good understanding of piezoelectric-based systems toward selecting suitable designs for potential aerospace applications and an outlook for novel designs in the near future.

## 1. Introduction

A concept of more/all-electric aircraft has recently received huge attention in the research and development work in the field of aerospace engineering [1,2,3,4,5,6,7]. The intent is to use more electrical systems in aircraft and aerospace applications to bring an impact on the environment [8]. With the fast development of electrification, more researchers and manufacturers are shifting to this dynamic trend involving a high demand for increasing the load, improving fuel efficiency, reducing emissions, and lowering the total cost of operation. Researchers seek different approaches and technologies to broaden this fashionable concept in a wide range of applications. The choice of actuators in the aircraft is based on various critical factors, such as power density, reliability, efficiency, control features, and thermal robustness, as well as the weight, size, and maintenance cost. In a commercial aircraft, actuators are essential in various applications, such as flight control, engine starter, landing system, brake actuation, and fuel pump [9,10]. The specifications of actuators in an aircraft vary across a wide range. Typical requirements can be listed as 1–320 kN of force, 10–700 mm of stroke, and 10–500 mm/s of speed, with the requirement of both modulated and two-position control methods [4,11]. For these actuation systems in the aircraft engine, the working temperature is from −50 to 150 °C at the engine intake; and it is higher for the actuators located toward the high-pressure compressor void (300–400 °C) or the tail cone area (500–600 °C) [12]. Overall, actuators in an aircraft require both the advantages of materials that allow them to deliver the required power in extreme environmental conditions and the optimal structural designs to maximize their performance within a constrained weight and space.

In the development of signal-by-wire and power-by-wire actuators in aircraft, electromechanical (EM) systems have seen a huge improvement, with significant results from both researchers and manufacturers. Electrical actuators, which have taken advantage of state-of-the-art motors and power screws, are among these systems [13,14,15,16,17]. The electrical actuators could provide a load range of up to 90 kN, with over 90% efficiency, making them suitable for replacing several conventional hydraulic or fueldraulic systems in the jet engine [18,19]. These systems bring more advantages in terms of a compact design (eliminating pipes and heavy elements) and power-to-weight ratio (weight reducing), enhancing aircraft stability and thus providing the ability to incorporate more functions within the control system to further enhance aircraft utility. Besides electrical actuators, smart-material-based actuators are also considered a promising approach. The development of smart materials, such as piezoelectric materials [20], shape memory alloys [21], magnetostrictive materials [22], and electroactive polymers [23,24], also offers advantages in the aerospace applications [25,26]. Looking beyond the potential of replacing the conventional system with similar or even better performance actuators, the smart behavior of such materials may offer more room for the development of novel systems. For example, the shape-changing ability of smart materials can be explored in morphing aircraft [27,28]. Shape memory alloy-based [29,30] and piezoelectric-based bender designs [31] can be used for noise reduction when mounting the bender on the trailing edge of the jet engine fan nozzle and the rotor of the helicopter, respectively. Each material responds differently to the stimuli, and various actuation modes can be achieved with distinct working concepts and geometrical designs. Among them, piezoelectric materials have shown great potential in aircraft and spacecraft applications [32,33,34,35,36]. The definition of piezoelectric materials is that they can either generate an output voltage when subjected to mechanical stress or perform a dimensional change when subjected to an electric field. These phenomena are known as direct and indirect modes of operation, which can be used for generators [37], sensors [38], and actuators [39]. Piezoelectric materials have the advantages of high power density, high efficiency, driving force, and displacement resolution over electromagnetic materials. They also do not generate electromagnetic noise and are nonflammable [40,41,42]. Piezoelectric materials come in different forms, such as sheet, wafer plate, stack, fiber, and composite, which makes them suitable for diverse geometrical designs. Despite a minimal strain capability, piezoelectric actuators can deliver high power outputs with high efficiency due to their ability to be cycled at very high frequencies as compared with other actuators [43] (Figure 1). 

Some piezoelectric materials can work in a very large temperature range, making them more promising in aircraft applications. A report from NASA revealed positive results of four piezoelectric ceramics, namely PZT-4, PZT-5A, PZT-5H, and PLZT-9/65/36, from several tests to evaluate their applicability as sensors and actuators in the intelligent aerospace system over a large temperature range, from −150 to 250 °C [44]. More efforts on material development were recorded that would gradually enhance the potential of piezoelectric materials in high-temperature industrial applications [45,46]. Therefore, piezoelectric-based systems are possible for applications located in the cold section of the aircraft engine, in which the temperature varies from −50 to 250 °C. However, amplification methods are required to generate sufficient stroke for these applications. The specifications of suitable applications for a compact piezoelectric design should be in the range of up to 5 kN of force, 100 mm of stroke, and 50 mm/s of speed. Thus the high stress and working frequency of piezoelectric actuators can be advantageous within a compact system. Some applications could be variable blow-in doors, booster bleeds, variable inlet guide vanes (IGVs), and variable stator vanes (VSVs). For instance, a piezoelectric-based linear actuator with a crank-slider mechanism was proposed to drive the IGV, which helps to control the flow that enters the jet engine and to improve the efficiency of the compressor [47,48]. Sufficient stroke of the actuator is accumulated over repeated cycles. For the same application (IGV or VSV of the gas turbine jet engine), the piezoelectric system could also be designed in such a way that a rotary motion can deliver directly to the application [49]. This actuator can be mounted on the unison ring, thus eliminating the need for other mechanical structures that add extra weight to the system. Moreover, the ability of power-off holding position of piezoelectric materials allows a design that can maintain the last controlled position in the event of failure, thus enhancing the safety level in aircraft applications.

This review paper focuses on the potential of piezoelectric-based systems for large-scale applications in the aircraft and aerospace industry. The structure of this review is as follows: Section 2 presents a brief overview of piezoelectric fundamentals, piezoelectric stacks, and classification of amplification methods. The subsequent four sections review the amplification methods for piezoelectric. Section 3 introduces direct amplification mechanisms to produce continuous motion. In Section 4, the quasi-static stepped actuators are reviewed and are divided into three concepts: inchworm, inertial, and walking. Section 5 reviews the ultrasonic actuators, where the resonant mode of piezoelectric is used. In Section 6, a different approach is described as the piezoelectric stack is coupled with hydraulic fluid in a pump to power the hydraulic cylinder. Section 7 summarizes the reviewed piezoelectric-based systems and discusses their potential in aerospace applications. Finally, the conclusion and future outlook are presented in the last section.

## 2. Piezoelectric Actuators

### 2.1. Fundamentals of Piezoelectric Materials

The piezoelectric effect on ceramic materials was discovered in 1880 by Nobel laureates Pierre and Jacques Curie. A piezoelectric transducer can be used as both generator [50] and actuator [51]. Specifically, the direct piezoelectric effect is used in the generator, while the indirect piezoelectric effect is used for the actuator [52]. The direct piezoelectric effect refers to the development of electrical charges on applications of mechanical stress, and vice versa (indirect piezoelectric effect). For the actuation applications reviewed in this paper, the piezoelectric material deforms with the applied electric field to produce mechanical energy.

The most commonly used piezoelectric materials are piezoelectric ceramic, such as lead zirconate titanate (PZT), barium titanate (BaTiO_3_), and lead titanate (PbTiO_3_). With a polycrystalline structure, ceramic materials can be fabricated into a variety of shapes and sizes. Besides, with the effort to reduce and avoid lead (Pb) in piezoelectric materials, lead-free piezoelectric development has been gaining momentum in recent years [20,53,54,55]. Some of these materials are alkali-metal-based bismuth sodium titanate (BNT), bismuth potassium titanate (BKT) [56], and potassium sodium niobate (KNN) [57]. To increase the potential of piezoelectricity in various working conditions, high-temperature piezoelectric materials have been developed, such as Pb(NbO_3_)_2_ and Bi_4_Ti_3_O_12_ [45]. However, the strain and stress of these materials may be reduced. In general, piezoelectric materials have a very small strain, of 0.1–0.2%, but with high stress, in the range of 100–131 MPa. Their specific power density is around 1000 kW/kg, and they have high efficiency, of more than 80% [40,43,58]. However, piezoelectric materials experience some drawbacks, such as substantial hysteresis [59], temperature-dependent properties [60], and fracture behaviors [61]. These phenomena eventually affect the performance of the piezoelectric materials, especially the stroke and accuracy of piezoelectric actuators.

The performance of the piezoelectric actuator is determined by the material properties known as the electromechanical coefficients. The most common material properties are the directional piezoelectric charge constants. The mechanical strain (S) of a piezoelectric material can be found by the relation
(1)$$S=s^{E}T+dE$$
where $s^{E}$ is the compliance or elasticity coefficient, T is the mechanical stress, d is the piezoelectric charge constant, and E is the electric field (E=Φ/t, where Φ is the applied voltage and t is the thickness of the material).

The coupling coefficient of the piezoelectric can be divided into three groups corresponding to the orientations of the electric field and the displacement. These coefficients are $d_{33}$, $d_{31}$, and $d_{15}$, corresponding to three deformation modes: longitudinal, transversal (Figure 2a), and shear modes (Figure 2b), respectively. In general, the strain and electromechanical conversion efficiency are higher in the longitudinal direction [62,63]. Therefore, this deformation mode is usually used in actuators, especially in the stacked configuration. Table 1 below shows examples of some piezoelectric materials and their properties that are commonly used in the piezoelectric-based system.

Piezoelectric elements can come in different geometrical forms, such as thin plate, single layer, multilayer, torsion tube. Besides these designs, macro piezo fiber composite (MFC) is another form of piezoelectric that was invented by NASA back in the 1990s and has been commercialized by Smart Material since 2002 [66,67]. Constructed of piezoelectric-ceramic-based fibers (usually PZT 5A or PZT Navy Type II) sandwiched between electrodes and polyimide layers, MFCs can produce elongation, contraction, and bending motions for actuation [68]. They can also function in sensitive sensor and vibration harvesting applications [69,70]. With a flexible nature, these piezoelectric composites have greater durability and reliability and can be attached to the surface of or embedded inside the structures. They have been proposed to be used in various aerospace applications, such as aircraft health structure monitoring, noise and vibration control of helicopter rotors, and surface control of morphing wings [67,69,71]. MFCs can find more aerospace applications if high-temperature piezoelectric (see Table 1), electrode, and adhesive materials are explored [72]. Table 2 below summarizes the performance of commercial piezoelectric actuators with some typical geometrical forms.

The piezoelectric actuator can be powered by a periodic voltage with sinusoidal, sawtooth, or rectangle waveforms. Depending on the required movements, each piezoelectric mechanism requires a customized input signal with a particular pattern, amplitude, and frequency to maximize its performance. In the event of more than one piezoelectric element being involved in the design, the phase difference of the controlled signal of each piezoelectric actuator needs to be designed precisely to obtain the coupling performance. The sinusoidal, square/rectangle, and sawtooth waveforms are commonly used in the stepped-motion piezoelectric system (Table 3). Power consumption during the operation of piezoelectric actuators is directly proportional to the capacitance of the device by the relation shown in Table 3.

### 2.2. Piezoelectric Stacks

Piezoelectric materials can be stacked together and be sandwiched between electrode layers to achieve a higher stroke for actuator applications [73,74]. Adopting the name of the manufacturing method, they are known as the piezoelectric stack or the multilayer piezoelectric (Figure 3). Piezoelectric stacks and piezoelectric actuators are manufactured and developed by various companies, such as Physik Instrumente (PI), Tokin Corporation, Cedrat Technologies, PiezoDrive, PiezoMotor, Piezosystem Jena, and CTS Corporation. Usually, the length of the stack is limited to 150 mm and the area is less than 225 mm^2^. The commercial piezoelectric stack usually offers a stroke range from several micrometers to a hundred micrometers (longitudinal mode) and a blocked force range from a hundred to a few thousand newtons. The size and shape of a piezoelectric stack can be customized to generate the required force and stroke. In the case of large stroke applications, an amplification mechanism is preferred.

The stroke (ΔL) of the stack is scaled with the number of stacking layers (Equation (2)), while the output force ($F_{b}$) is related to the active area of the piezoelectric actuators (Equation (3)).
(2)$$\Delta L=V_{pp}\times d_{33}\times N$$
(3)$$F_{b}=V_{pp}\times d_{33}\times YA/L_{0}$$
where, $d_{33}$ is the piezoelectric constant (longitudinal mode), N is the number of stacking layers, Y is the modulus of the piezoelectric material, A is the area of each layer, and $L_{0}$ is the initial thickness of each layer. The force and stroke of the piezoelectric stack are under an electrical load (applied voltage $V_{pp}$), and the mechanical load is shown in Figure 4. When the piezoelectric stacks are implemented in a cyclic process, they will be subjected to a severe hysteresis characteristic affected by the frequency and magnitude of the applied voltage. Therefore, closed-loop control is required to compensate for the hysteresis effect in precise positioning applications [59,75].

### 2.3. Classification of Amplification Methods

The microstroke range of a stand-alone piezoelectric stack can be further amplified to the required level using external amplifiers, such as mechanical, hydraulic, or other kinetic mechanisms, depending on the architecture. Several conceptual designs are proposed and used, such as the amplified mechanism by the compliant structure, the inchworm mechanism, the walking mechanism, and the hybrid electro-hydraulic system. In this paper, the amplification methods are divided into four groups, as shown in Figure 5. Each technique will be discussed in subsequent sections.

## 3. Continuous Motion

The small stroke can be amplified instantly once the piezoelectric element is activated. Compliant mechanisms, cantilever, X-frame mechanism, and unimorph/bimorph configurations could be classified under this group. They are capable of generating a smooth and continuous motion with less friction and zero backlash in a compact design. However, there are trade-offs between the output force, the overall stiffness, and the response speed for the displacement. The output stroke is defined by the input stroke from the piezoelectric element and the amplification ratio of the amplifier. The amplification ratio is usually limited to tens due to geometrical constraints. Besides, the relative size of the piezoelectric element should not be too large as it would lead to an oversized system.

These amplification methods can generate a moderate stroke or increase the input stroke from the piezoelectric element for other designs. The commercial amplified piezoelectric actuator of Cedrat Technologies company, for example, could generate a stroke of up to 1 mm. This stroke is 10 times or more larger than that of a stand-alone piezoelectric stack in a load-free condition [33,76]. For a broader stroke range, the stepped-motion mechanisms would offer better solutions [39].

### 3.1. Compliant Mechanism

The micron stroke of a piezoelectric stack can be amplified by up to thousands of micrometers by a compliant mechanism [77,78,79]. These mechanisms can be constructed by rigid arms with flexure hinges (Figure 6a) or by thin arms (Figure 6b). The piezoelectric stack applies a mechanical force to the system and causes the elastic deformation of the compliant mechanism. As a result, the output motion is generated with an amplification ratio of ΔY/ΔX either in a perpendicular direction to (Figure 6) or in the same direction (Figure 7) as the piezoelectric stroke. On amplifying the stroke, the output force decreases.

The amplifying ratio can also be maximized by modifying the structure of compliant mechanisms. Several designs were proposed and used in applications, such as bridge type [80,81], rhombus type [78,82], Scott Russell type [83,84], and honeycomb type [85]. The amplified piezoelectric actuator can also be used to enhance the input stroke. An optimal geometrical design can be achieved to provide a specific stroke and force. The designs with compliant mechanisms usually offer an amplification ratio of up to a few tens.

Besides the bridge type, which is commonly seen in the commercial amplified piezoelectric actuators (Figure 6), the lever mechanism offers simplicity in design, manufacture, and assembly among amplification methods [86,87]. A simple lever mechanism [42] (Figure 7a) can also be modified to an X-frame mechanism/scissor mechanism [88] (Figure 7b). The output motion is parallel to the stroke input from the piezoelectric stack. Similar to other compliant mechanisms, the lever mechanism trades force for stroke amplification. A design of two-stage cantilever mechanism can have an amplification factor of 30, with a final stroke of 400 μm, to be used as a printed head [62]. Another mechanism, the so-called high-bending-stiffness connector, which was recently proposed and commercialized (in 2018), can amplify the output stroke to by two–three times, with only a fractional increment in length [89]. These compliant mechanisms can also be integrated into the design to gain a long input stroke for other actuators, such as inchworm [90,91], inertial [92,93,94], and piezoelectric-hydraulic pump [86].

### 3.2. Piezoelectric Bending

The piezoelectric plate can be used to create bending motion for various applications [95,96,97]. Figure 8a illustrates a schematic of a unimorph configuration with one piezoelectric plate and one passive plate. In other applications, a bimorph configuration can be constructed from two piezoelectric elements to create a two-way bending motion (Figure 8b).

The bending motion from unimorph and bimorph configurations can be used in the inertial piezoelectric actuator [98] or as an active valve of the piezoelectric micropump [99]. A combination of a piezoelectric bender and a compliant mechanism was also proposed to control a helicopter rotor blade [100].

The amplification methods in this section (Section 3) are usually considered in those applications where the stroke is the priority and space is not constrained. For applications where a higher stroke is required, other amplifiers can be selected for a compact design.

## 4. Quasi-Static Stepped Motion

An output stroke of up to centimeters could not be achieved directly from the amplification methods covered in Section 3. It requires a further cumulative effect whereby the stroke can be accumulated from microstep motion over repeated cycles. These mechanisms can be classified as inertial, inchworm, and walking concepts. Theoretically, these amplification concepts could produce unlimited output motion. However, in practice, the piezoelectric-based actuators using such methods are designed to have the output motion in the range of centimeters, with a maximum speed of tens of millimeters per second. For more significant stroke requirements, these stepped-motion actuators could be scaled up. However, they require a suitable housing structure to provide the backbone and support. This housing structure must not be oversized as compared to the actuator. As the working principle is mostly based on friction, they require tight tolerance of the structural dimension and the frictional holder. For high-force applications, they are prone to mechanical wear and tear over time. Thus, regular maintenance of such systems is required. The piezoelectric stack is usually driven at a low frequency, of less than 1 kHz. Therefore, these actuators can also be considered as a quasi-static system and are different from the ultrasonic system discussed in Section 5.

### 4.1. Inertial Concept

The inertial actuator is built from one moving block (piezoelectric stacks), fixed at one end, and one inertial mass acting as a friction element (Figure 9). The working principle of the inertial piezoelectric-based actuator can be described as follows: First, the piezoelectric stack is controlled to extend slowly. During this process, the friction element makes contact with the moving structure; hence, they are moving forward together due to the frictional force. After that, the piezoelectric stack contracts quickly to create an impulsive force. Due to inertia, the moving structure cannot respond to the fast retraction to return to its original position but remains in its current place. As a result, the moving block returns to get ready for a new cycle while the moving structure is brought forward. The piezoelectric stack is powered by a sawtooth wave input signal to create the required motion. This design can be used for long-stroke and high-resolution applications [101,102] by slowly increasing voltage to drive the piezoelectric stack.

The concept demonstrated in Figure 9 can also be called the stick-and-slip mechanism. The roles of piezoelectric stack, friction element, and moving structure are exchangeable [92,103,104,105]. In the impact drive mechanism, the position of the moving unit (piezoelectric stack and friction element) changes in each step, while the moving structure is now fixed [106]. Flexure hinges can be introduced to generate the required motion from piezoelectric elements [92,93,94,104]. Moreover, these mechanisms also increase the effective displacement of the piezoelectric element in the designed direction. Table 4 summarizes some inertial piezoelectric actuators with both symmetrical and asymmetrical flexure hinges. A design with one piezoelectric stack and asymmetrical flexure hinge can produce a linear motion with a speed of up to 15 mm/s [93].

### 4.2. Inchworm Concept

The piezoelectric inchworm creates stepped motion by mimicking the crawling motion of an inchworm [91,108]. Figure 10a shows the basic principle of the inchworm design with three sets of stacked piezoelectric actuators: a moving block (2) is extended and contracted to provide the main motion to drive the moving structure, while two clamping blocks (1 and 3) are engaged and disengaged with the moving structure one at the time. With this geometrical relation, the piezoelectric inchworm allows a straightforward design for each piezoelectric block to achieve the required level of pushing and clamping force.

The working principle of the inchworm actuator can be explained as follows: First, clamping stack 3 extends and clamps the moving structure below it. The moving stack (stack 2) then extends and pushes stack 3 to move forward (toward the right side) together with the moving structure. Next, clamping stack 1 extends down to clamp the moving structure while stack 3 retracts. Stack 2 contracts to drive stack 1 and the moving structure further toward the right side. After that, stack 1 retracts to release the structure while stack 3 extends to engage again. These steps are repeated to accumulate small steps into significant motion. In this design, the inchworm unit (three piezoelectric stacks) remains the same while the moving structure is being pushed in one direction only. Therefore, this design can be called a pusher inchworm. In another modification, called the walking inchworm, the moving structure is now fixed while the inchworm unit moves, which resembles the inchworm crawling on the tree branch. The clamping and moving blocks in inchworm piezoelectric can be divided into several blocks and arranged along with the moving structure, as shown in Figure 10b These piezoelectric blocks can be controlled to achieve the desired performance.

Several works are geared toward developing the inchworm design from a geometrical design to a control method for various targeted applications [39,108,109,110,111,112,113]. The clamping force can be achieved by either the intermittent [114,115,116] or the continuous mechanism [113]. In the intermittent clamping mechanism, the clamping blocks provide the clamping force to the structure in sequence. In contrast, the continuous clamping mechanism maintains contact with the structure during the working process. Therefore, the clamping force varies and depends on the load condition for the intermittent mechanism, while the continuous mechanism can only generate a constant clamping force. The clamping structure can be a piezoelectric stack [108,115,116], electromagnetic in nature [49,117], an inertial mass [118,119], or a wedge-type clamping mechanism [111,120]. To increase the velocity of the motion, the stroke of the moving block can be amplified using a flexure mechanism, as mentioned earlier, in Section 3.1 [90,91]. The moving block can also be replaced by a magnetostrictive actuation with a similar or larger stroke (0.2% strain). An inchworm actuator with Terfenol-D as a moving block and piezoelectric stacks as a clamping block can generate a stall load of 115 N and a no-load speed of 2.5 cm/s [121].

### 4.3. Walking Concept

The main difference between the walking concept and the inchworm concept is that the walking type does not require a moving block. Instead, the moving motion is created directly from the walking legs. The legs produce an elliptical motion on their tips to engage and disengage with the moving structure in sequence to drive it further. In the walking concept, the position of the leg structure remains the same. The number of legs can vary depending on the design architecture, with a minimum of two legs [122].

As shown in Figure 11, the working principle of the walking concept is described as follows: Legs 1 and 3 (leg group I) come in contact with the moving structure and bend to the right to drive it forward while legs 2 and 4 (leg group II) retract and bend to the left. Next, leg group II extends to make contact with the moving structure; then, they bend to the right to drive the moving structure further. Leg group I retracts and bends to the left to repeat the previous motion of leg group II. These steps are repeated in sequence to create stepped motions of the moving structure.

The bending movement of the leg could be created by piezoelectric in bending mode [123] (Figure 12a), V-shape configuration [124] (Figure 12b), or combining the longitudinal and shear motions [125] (Figure 12c). The output force and speeds of these designs are compared in Table 5. Several products using walking concepts are commercialized for small-scale applications with the force range of around a few hundred newtons, a designed stroke of less than 100 mm, and a speed of less than 20 mm/s [126,127]. A series of PiezoWalk from Physik Instrumente company, for example, could produce a linear motion with a velocity of 15 mm/s (PICMAWalk) or a maximum blocking force of 300 N [126].

Similar to the other stepped-motion design, flexure hinges can also be employed to create the bending motion for the walking leg from only one piezoelectric stack. This could reduce the number of piezoelectric elements and the input signal of the system. For example, two asymmetrical right-angle flexure hinges could generate a linear motion with a motion speed of 39.78 μm/s at a frequency of 20 Hz [128].

## 5. Ultrasonic Stepped Motion

While the quasi-static stepped-motion designs operate at a low frequency, ultrasonic actuators use piezoelectric resonant vibrations [129]. Therefore, they are capable of producing high velocities and long-range motions. The ultrasonic actuator consists of a piezoelectric-based stator and a moving structure (Figure 13). The stator produces the elliptic motion to drive the moving structure that is similar to the piezoelectric walking design. The elliptic motion could be generated from the tip of each piezoelectric element or the combination motions of the whole stator [130]. Based on these driving methods, the ultrasonic actuator can be classified into two groups: standing wave [131,132,133] (Figure 13a) and traveling wave [39,134,135] (Figure 13b). The ultrasonic concept could be used for both linear motion [136] and rotary motion [135,137].

## 6. Hybrid Piezoelectric–Hydraulic System

The piezoelectric mechanisms described in previous sections always require a trade-off between force and stroke. Therefore, hydraulic energy may be more realistic for applications requiring both massive stroke and force (in the range of a thousand newtons and more). In such cases, hybrid piezoelectric-hydraulic systems would be beneficial. For example, a piezoelectric actuator with a higher energy density and lower power consumption than the electromagnetic actuator would make it a promising candidate for the pump in an electro-hydrostatic actuator [33]. Hybrid piezoelectric–hydraulic actuators have been researched and used in various fields, such as aerospace [138], automotive [139], and mechanical engineering [140]. The basic concept lies in coupling a piezoelectric stack with the transmission of hydraulic fluid via valve systems. The high frequency, large force, and small stroke of the piezoelectric actuator can be converted into a lower frequency and larger stroke of the output cylinder [141,142,143,144,145,146,147] (Figure 14a). Piezoelectric stacks can be operated by hundreds of volts, making it suitable for available electric power in aircraft. Unlike the conventional hydraulic system in the jet engine, the hybrid piezoelectric-hydraulic system only consumes power when required to move load. Therefore, it would have high energy efficiency. Besides, the piezoelectric-hydraulic pump has few moving parts; it also eliminates the need for lubrication, hence reducing the maintenance effort.

The working principle of the piezoelectric–hydraulic pump is shown in Figure 14b. It is described as follows: When the piezoelectric stack expands, the piston is pushed further, thereby decreasing the volume of the pump chamber. Pressure is built up inside the pump chamber, causing the outlet valve to open to release the fluid. When the stack contracts, the volume of the pump chamber increases. The decrease in chamber pressure causes the inlet valve to open, allowing fluid to enter the chamber. This process is repeated to control the fluid flow and regulate the chamber pressure from the piezoelectric-hydraulic pump.

The piezoelectric–hydraulic pump has advantages over conventional systems such as hydraulic, pneumatic, and electric actuators in terms of efficiency and power density [139]. Its performance can be characterized by the maximum flow rate of the working fluid and the stall pressure [148]. This fluid circuit will then be extended into a working cylinder via a control valve system and accumulator. Some researchers have worked on developing the pump design to maximize the flow rate and pressure performance [144,149,150,151,152,153] while other researchers have attempted to build a compact system with an integrated hydraulic cylinder [139,150,154,155]. The performance of this hybrid pump relies on both the design of the piezoelectric stack and the pump chamber and the use of hydraulic fluid and the valve system.

### 6.1. Design of the Piezoelectric Configuration

The first publication on the piezoelectric–hydraulic system proposed the use of a piezoelectric stack to drive the hydraulic circuit [149,156]. A piezoelectric stack 55.5 mm in length and 22 mm (stroke of 60 μm) in diameter is driven from −100 V to 500 V at a frequency of 300 Hz and could produce an output power of 34 W. Based on this result, several improvements have been proposed. To increase the performance of the pump, the frequency of the piezoelectric stack in the system was investigated. It was found that the piezoelectric stack self-heating phenomenon is one of the critical issues that occurs in PZT materials [141,157]. Therefore, a maximum pumping frequency of 1 kHz was selected based on the thermal limitations of piezoelectric stacks [154]. For higher-frequency applications, a cooling system was introduced to enhance the performance of the piezoelectric stack. The cooling fluid was introduced so that it could be used to thermally regulate the piezoelectric stack operating at a high frequency [140,158].

Research on a compact piezoelectric–hydraulic system was conducted at the University of Maryland [150,154,155]. At a high frequency, significant losses in flow rate were observed. It demonstrated a highly nonlinear variable of the output velocity with pumping frequency. By comparing the performances of the hybrid pumps from different piezoelectric materials [159], they found that higher power output could be achieved from single-crystal PMN-based materials. Besides piezoelectric materials, some hybrid pumps also use magnetostrictive materials, such as Terfenol-D and Galfenol [138,155,160].

### 6.2. Design of the Pump Chamber

The size ($A_{\mathrm{piston}/\mathrm{chamber}}$) of the piston and the pump chamber is based on the performance of the piezoelectric stack ($Stroke_{\mathrm{piezo}}$, $Force_{\mathrm{piezo}}$, and Frequency) and pump performance (Flow rate and $Pressure_{\mathrm{chamber}}$) (Equation (4)):(4)$$A_{\mathrm{piston}/\mathrm{chamber}}=\frac{Flow\;rate}{Stroke_{\mathrm{piezo}}\times Frequency}=\frac{Force_{\mathrm{piezo}}}{Pressure_{\mathrm{chamber}}}$$

The design of the multipump chamber adopted from micropump designs [99,161,162] can be investigated in the piezoelectric stack pump to increase its performance. The double-piezoelectric pump was reported to have a significant increase in both the flow rate and the output pressure and produce a continuous fluid flow inside the system [163].

The sealing method is also an essential issue in the pump chamber design to prevent fluid leakage. The pump chamber can be sealed by O-rings arranged on the long side piston or a thin diaphragm plate. The first method is used in the pump design with a large input stroke [164]. The second one is commonly used in the design with a small to medium input stroke [147,158,165,166,167]. However, a thin diaphragm plate will be deformed permanently after repeated cycles. It can lead to a reduction in pump performance. The working fluid selection is also important to reduce the leakage and increase the pressure generated within the chamber. Some working fluids used in the piezoelectric–hydraulic pump are the hydraulic fluid MIL-H5606F [154], the water-based hydraulic fluid Hydrolubric 123 [168], Mobil DTE-24 [163], glycerin solution [169], and AeroShell oils [138]. The ionic liquid (IL) with a higher bulk modulus than other common working fluids was also proposed to increase the output pressure [170]. Besides, the operating temperature range of the fluid is also an essential criterion for fluid selection. They should be sufficient for aerospace applications in which the ambient temperature may vary across a broad range, from −50 to more than 250 °C. Therefore, in some cases, thermal solutions are important to assure the stable performance of the system. The properties of typical working fluids are shown in Table 6, below.

### 6.3. Design of the Valve System

The valve system is important to determine the performance of the piezoelectric–hydraulic pump. The valves are designed to regulate the fluid flowing into and out of the pump chamber. Therefore, the response of the valves needs to be compatible with the movement of the piston (or piezoelectric stack). The valves used in piezoelectric–hydraulic pumps are reed valve, microreed valve, active valve, and diffuser valve (valve less design).

The reed valve is popular among all valve types (Figure 15a). The reed valve structure is simple, and its movement is passively related to the piezoelectric stack’s performance. However, the response of the reed valve is limited by the natural frequency of the geometrical design. While the piezoelectric stack can be operated at hundreds to thousands of hertz, the reed valve movement is usually at around a few hundred hertz, depending on the geometry and size. A miniaturized piezo-hydraulic pump was developed with the highest frequency of the reed valve at 400 Hz [176]. Other attempts to change the design of the reed valve with the microarray valves have focused on increasing the frequency but still maintaining the working condition of the pump. The microvalve arrays with a spider-spring (or arm) configuration [158,168,177] (Figure 15b) could enhance the performance of the pump function, especially the flow rates.

The active valve is formed by a unimorph disc type of piezoelectric for fluid flow rectification [165]. This active valve allows it to open more rapidly than the reed valve, as well as it can reduce flow resistance. Backflow can also be suppressed. By controlling the valve operation corresponding to the movement of the piezoelectric stack, the delivered fluid volume could be maximized. However, existing research has reported that the actual flow rate is lower than the expected value due to the appearance of air entrapment and the effective control of the active valve. Later research has shown the importance of timing control of the active valve in the performance of the system at a high frequency, which was missing in the previous study [178].

The diffuser valve or valveless design is also used in the piezoelectric pump, but mostly in a micropump driven by a diaphragm piezoelectric. This valve type can be conical [179,180] or a Tesla valve [153,181]. These designs do not have any moving parts, so they do not suffer from fatigue failure as compared to other valves. This valveless structure is much smaller than designs with valves as it does not have a flow rectification system. Its performance relies on fluid flow from a high-pressure source to a low-pressure place.

Table 7 Summarizes several valve types in the piezoelectric–hydraulic pump. Each valve has its pros and cons that can still be developed to maximize the performance of the piezoelectric pump. A control strategy can be created for the active valve to synchronize its movement with the pump function. Finally, the concept of the diffuser valves can be integrated with a piezoelectric-stack pump.

The performances of some piezoelectric–hydraulic pumps are presented in Table 8. The experimental results are based on prototypes with a maximum piezoelectric input stroke of less than 100 μm. The reported flow rates were less than 2 L/min, with the stall pressure of a few thousand kilopascals. For practical applications, a piezoelectric–hydraulic pump can be scaled up to increase its performance. The most powerful commercial piezoelectric-based pumps are the solid-state pumps SSP from Kinetic Ceramics company, with reported pressure in a stalled condition of 2700 psi (18.6 MPa) and a maximum flow rate of up to 7 L/min, using high-voltage piezoelectric stacks (operating voltage 0–1000 V) working at 1 kHz of frequency [164].

## 7. Piezoelectric-Based Systems in Aerospace Applications

Harnessing various amplification methods listed earlier, the potential of piezoelectric actuators can be further exploited significantly. Each design has a different performance range (force, stroke, resolution, speed) and requires different structural designs and control strategies. Table 9, below, shows a comparison of these piezoelectric-based systems studied in this paper.

The designs of continuous motion are the simplest to directly amplify the stroke of the piezoelectric by a certain ratio. Amongst them, the flexure hinge is the most popular mechanism that is used in both research prototypes and commercial products. The amplification ratio can be adjusted by the geometrical design but is usually limited to a few tens. Commercial products usually produce output strokes in the millimeter range and an output force of thousands of newtons. These structures can be used directly for applications with requirements in this range or in combination with other mechanisms [33].

For broader stroke application, the stepped-motion systems are preferred. The output stroke is achieved by accumulating small motions after steps. The resolution, speed, and force abilities vary in each concept. Generally, quasi-static systems are more suitable for larger force and slower speed applications compared to ultrasonic systems. These methods are used in various commercial piezoelectric drives with the designed stroke length in the centimeter range. The available output force and speed are tens to hundreds of newtons and a few to hundreds of milimeters/second, respectively. These systems can be scaled up to achieve larger outputs, but they are limited due to the space constraint of the applications.

The performances of some commercial piezoelectric products are summarized in Figure 16 (from Physik Instrumente (NEXLINE, PICMAWalk, NEXAC-walking concept; Inertia Drives; PILine Ultrasonic) [126], PiezoMotor (LEGS linear-walking concept) [127], Cedrat Technologies (APA-continuous motion) [76]). The output force varies in each product based on the size of the piezoelectric elements. Stepped-motion actuators could provide a maximum force of a few to hundreds of newtons. Amplified piezoelectric has a maximum force of hundreds to a few thousand newtons. The standard piezoelectric stacks are usually manufactured to produce a force in a range of hundreds to thousands of newtons. High-force actuators can be achieved by combining a number of piezoelectric stacks to increase the effective cross-sectional area. The stroke of each piezoelectric stack is small, in the microrange. However, by using the stepped-motion concepts, the stroke of piezoelectric devices can be increased up to the centimeter range. The ability to be operated with a high frequency (ultrasonic devices) allows the design to gain centimeters/second of output velocity. These performances are sufficient for those aircraft applications as mentioned in Section 1. Furthermore, with a resolution of up to micro- and nanorange, piezoelectric devices could add an extra advantage in precise positioning applications if required. In applications in which multiple sets of the stepped-motion piezoelectric systems are used, the resolution of each piezoelectric system plays a key role in the control method. Thus synchronizing individual performances of piezoelectric sets helps gain high output force and stroke.

The piezoelectric-hydraulic pump is another approach to amplifying the performance of the piezoelectric by coupling it with a working fluid. This pump, with a significant flow rate and stalled pressure, has been commercialized by Kinetic Ceramics company [164]. Several products from the solid-state pump series SSP can generate output pressure of 100 to 2700 psi (~0.69 to 18.6 MPa) and a maximum flow rate of up to 7.5 L/min (Figure 17). These pumps can work with various working fluids and in extreme environmental conditions (−40 to 125 °C), which makes it possible to replace the conventional hydraulic or pneumatic pump in aerospace applications.

Considering the potential of piezoelectric-based systems, it is possible that such systems could be developed further and integrated into various aerospace applications. The selection of a conceptual design depends on the stroke and load range. For example, the bridge-type piezoelectric actuator has also been used to build a tip-tilt mechanism for micron positioning or a linear stepping actuator using a stick-slip concept [183]. In other applications in the helicopter, a bimorph piezoelectric actuator constructed by two piezoelectric ceramic plates was used to deflect a trailing edge flap on the rotor blade [100] (Figure 18b). To achieve a large stroke, a piezoelectric-stack-based system was proposed in the design where micron strokes are accumulated by using a feed-screw for morphing aircraft structures [184]. Piezoelectric stacks are also proposed to be embedded in the trailing-edge flap in the main rotor of the MD900 helicopter. This system allows active control of the flap, thus improving the aerodynamic performance and reducing the vibration, noise, and power consumption of the rotor [74,185]. Progressively, amplified-piezoelectric actuators with compliant mechanisms have been used for the active flap (Figure 18a) [28,31,183]. In another stepped-motion concept, a linear inchworm piezoelectric actuator has been proposed for positioning engine inlet guide vanes via a crank slider mechanism [47,48] (Figure 18c). The inchworm concept can be used to generate rotary motion to directly drive the unison ring to control the inlet guide vanes, hence reducing the transmission mechanism from the previous linear actuator [49]. In the approach of the hybrid hydraulic system, the piezoelectric pump developed by Kinetic Ceramics Inc. has been tested with hydraulic primary flight control in remotely piloted vehicles [158,168]. The solid-state pumps from this manufacturer have been improved in terms of performance over the years for more practical applications. Following the same working concept as that of electro-fluidic components with smart materials, piezoelectric stacks can be used as precisely controlled valves for a magnetostrictive pump [138].

However, some limitations may need to be considered when using piezoelectric-based systems. First, material aging causes a change in the properties of materials and loss of polarization, which results in instability over a long working period. Second, the temperature dependence of properties limits the working condition of the piezoelectric systems. Hence, piezoelectric-based systems may be suitable for such applications exposed to ambient temperature in a range of −50–150 °C. In the case of a higher working temperature, a thermal solution is required. Moreover, as a brittle material, piezoelectric is prone to be easily damaged by tension. Therefore, it requires careful design and operation to suppress unexpected tensile loads. Finally, the enclosure may be vital for stepped-motion actuators to eliminate the working environment’s effect on the friction elements. Knowing these limitations, the development of futuristic piezoelectric materials is essential besides the conceptual designs. Piezoelectric materials could be tailored to alter their material properties to be better suited for aerospace applications.

## 8. Conclusions

Piezoelectric-based systems can be considered as novel electromechanical designs to use in aerospace applications, especially toward the concept of more electric aircraft. To increase the potential of piezoelectric systems, several amplification methods and related conceptual designs have been reviewed in this paper. Understanding the mechanisms and their specifications is beneficial to technology selection. These mechanisms can be divided into four amplification groups: continuous-motion, quasi-static stepped-motion, ultrasonic stepped-motion, and piezoelectric–hydraulic systems. The designs in the first three groups can directly generate output force and stroke, while the piezoelectric pump reviewed in the last group produces a fluid flow to power the hydraulic cylinder. Moreover, continuous motion from the first group can be used to enhance the input stroke from the piezoelectric element for other systems.

Even though most of the current research prototypes and commercial products based on piezoelectric serve in small-scale applications with moderate force and stroke ranges, the concepts of stepped motion and piezoelectric–hydraulic have the potential to be scaled up and developed for large-scale applications. Some examples of aerospace applications and developmental products have been introduced to highlight these possibilities. However, to scale the actuators for practical applications, new challenges must be overcome. Manufacturing and assembly are the most critical issues for those concepts using frictional elements. For a large prototype, the housing structure needs to be optimized to avoid having an oversized system. Besides, some properties of piezoelectric materials, such as hysteresis characteristics, temperature-dependent properties, or aging, may reduce the overall performance of the systems. Therefore, they need to be considered in the overall design and control process; else, regular maintenance is required.

## Figures and Tables

**Figure 1 micromachines-12-00140-f001:**
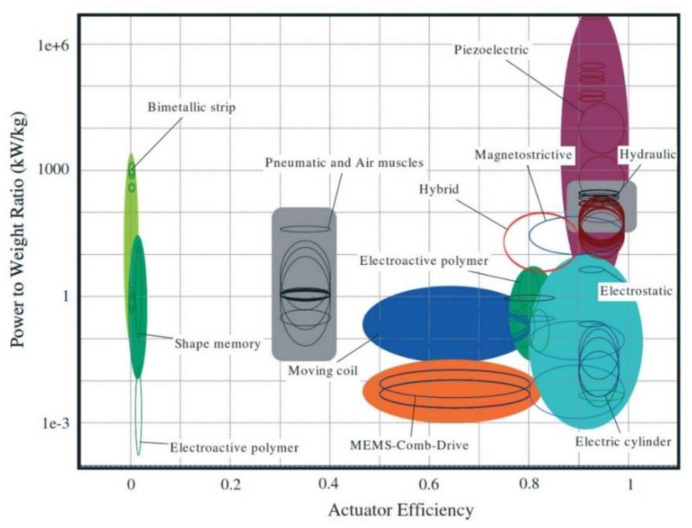
Power-to-weight ratio versus efficiency from a database of 220 actuators (from [43]).

**Figure 2 micromachines-12-00140-f002:**
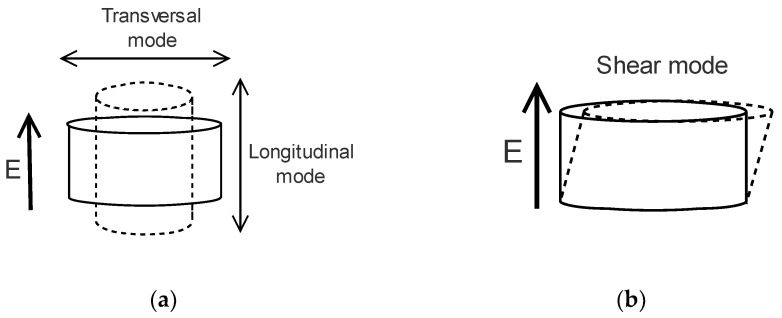
Deformation of the piezoelectric actuator. (**a**) Longitudinal and transversal mode; (**b**) shear mode.

**Figure 3 micromachines-12-00140-f003:**
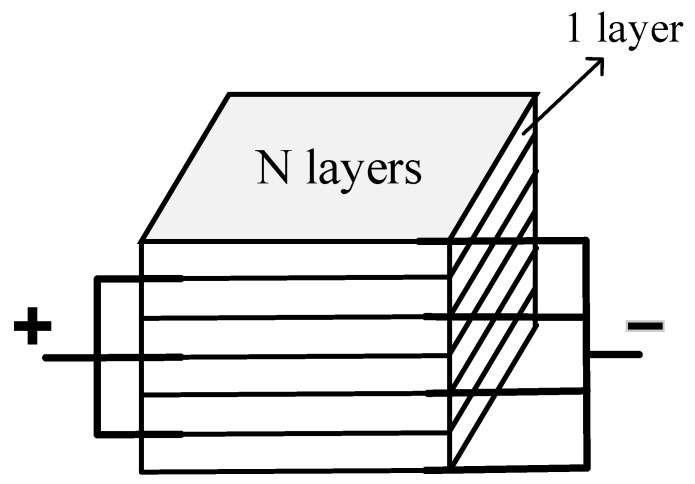
The electrical connection of an *N*-layers piezoelectric stack.

**Figure 4 micromachines-12-00140-f004:**
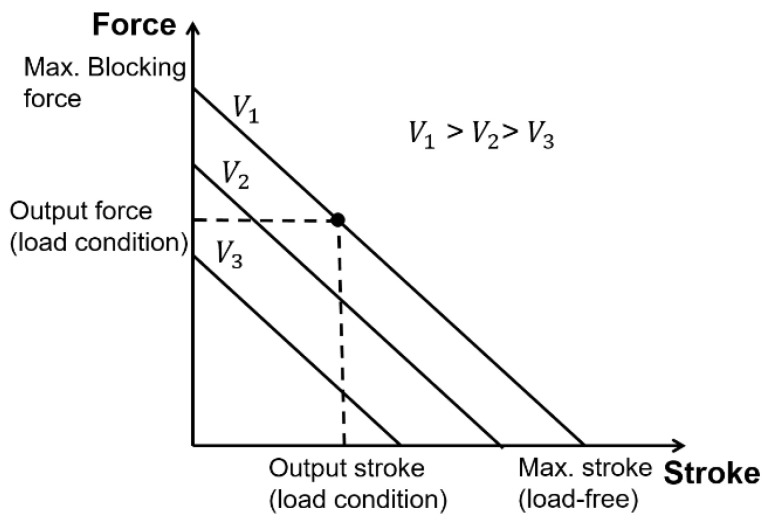
Force–stroke characteristic of the piezoelectric stack.

**Figure 5 micromachines-12-00140-f005:**
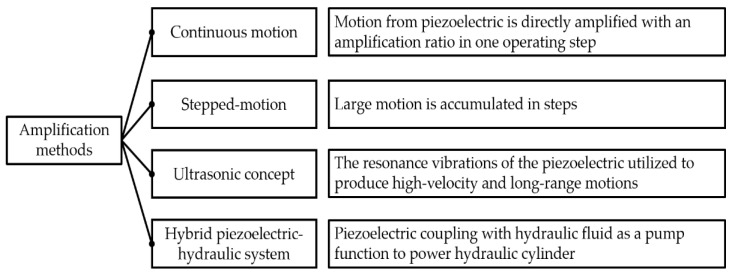
Summary of amplification methods.

**Figure 6 micromachines-12-00140-f006:**
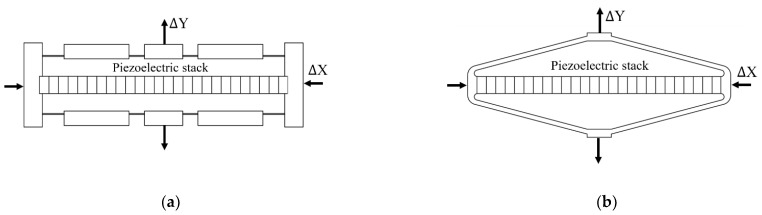
The output is perpendicular to the input: (**a**) rhombus type with rigid arms and flexure hinge; (**b**) bridge type with thin arms.

**Figure 7 micromachines-12-00140-f007:**
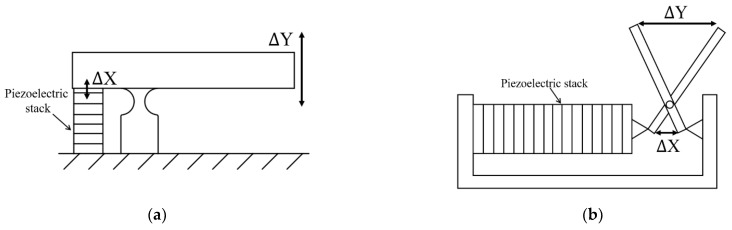
The output is in the same direction as the input: (**a**) lever mechanism with a rigid arm and a flexure hinge; (**b**) X-frame/scissors mechanism.

**Figure 8 micromachines-12-00140-f008:**
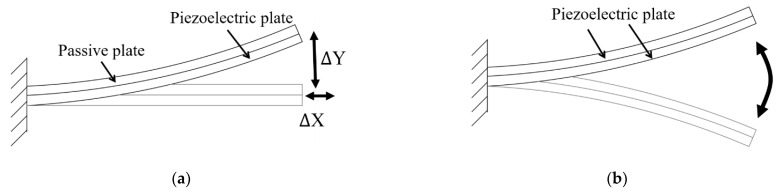
Schematic of (**a**) unimorph configuration and (**b**) bimorph configuration.

**Figure 9 micromachines-12-00140-f009:**
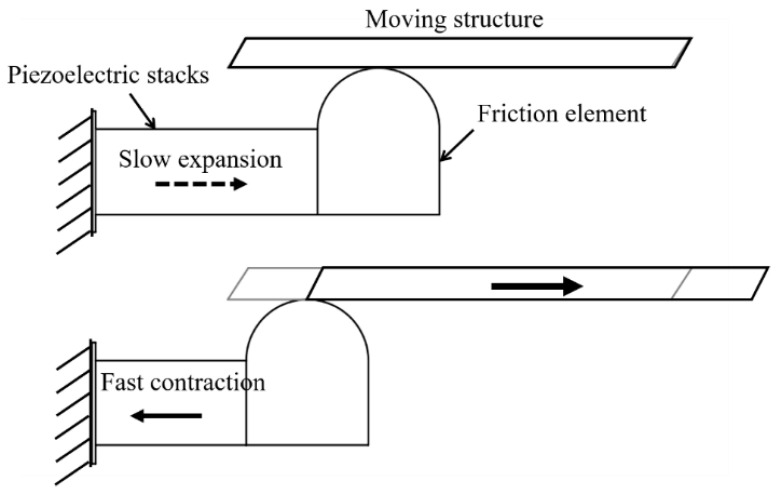
Schematic of the inertial piezoelectric actuator.

**Figure 10 micromachines-12-00140-f010:**
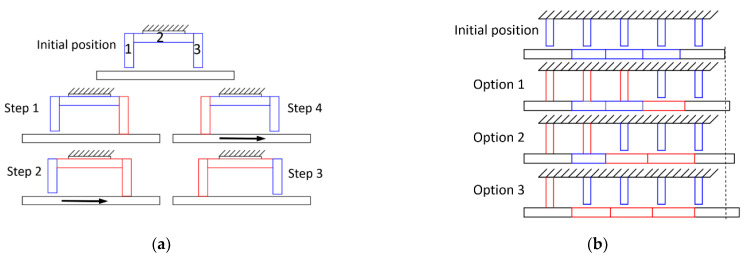
Schematic of the inchworm mechanism with stacked piezoelectric actuators. (**a**) Standard inchworm system with three piezoelectric stacks: stacks 1 and 3 are to create a clamping force, and stack 2 is to generate the movement. (**b**) Improvement of the inchworm concept with various clamping blocks and moving blocks integrating into the moving structure. Blue block: inactive piezoelectric (contract); red block: active piezoelectric (extend); black block: moving structure.

**Figure 11 micromachines-12-00140-f011:**
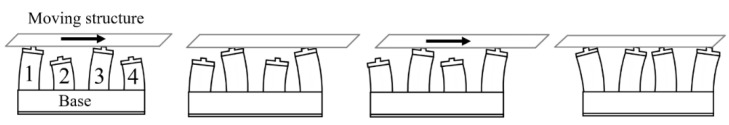
Schematic of a walking piezoelectric actuator.

**Figure 12 micromachines-12-00140-f012:**
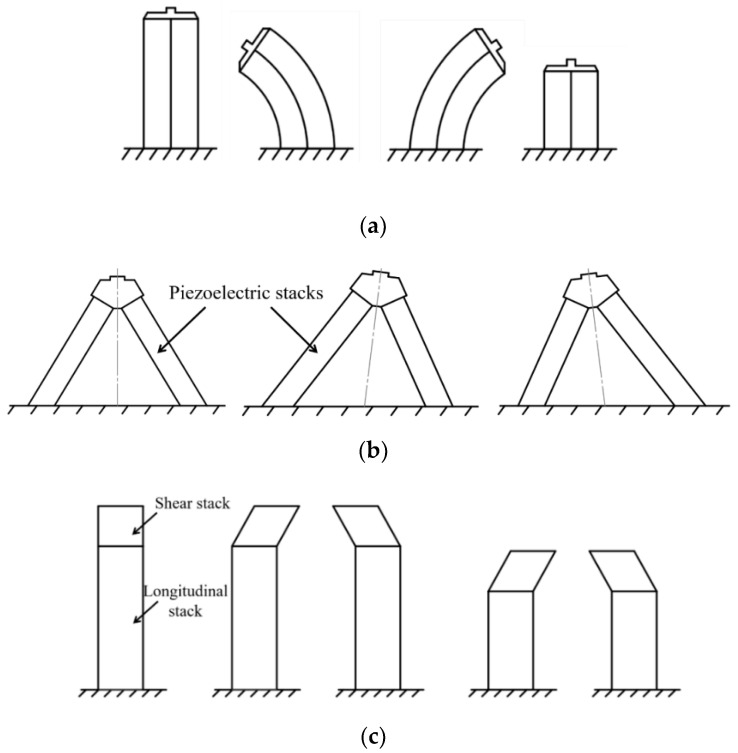
Designs of walking legs: (**a**) bimorph configuration; (**b**) V-shape configuration; (**c**) combination of longitudinal and shear modes of piezoelectric.

**Figure 13 micromachines-12-00140-f013:**
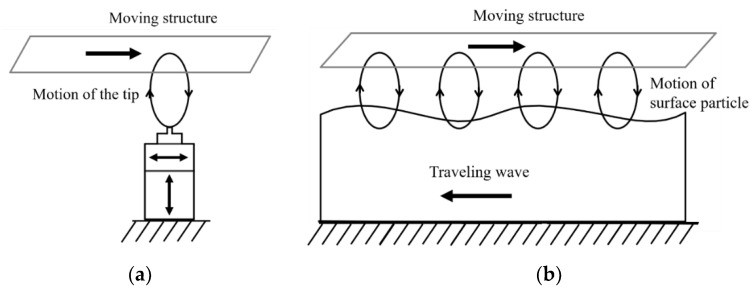
Working principle of the piezoelectric-based ultrasonic actuators: (**a**) standing wave; (**b**) traveling wave.

**Figure 14 micromachines-12-00140-f014:**
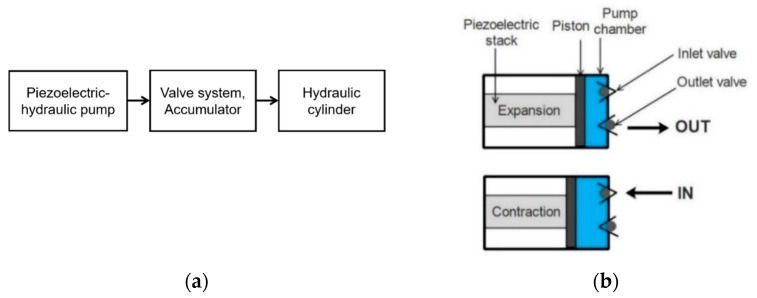
(**a**) Flowchart of the hybrid piezoelectric–hydraulic system. (**b**) Working principle of the piezoelectric-hydraulic pump.

**Figure 15 micromachines-12-00140-f015:**
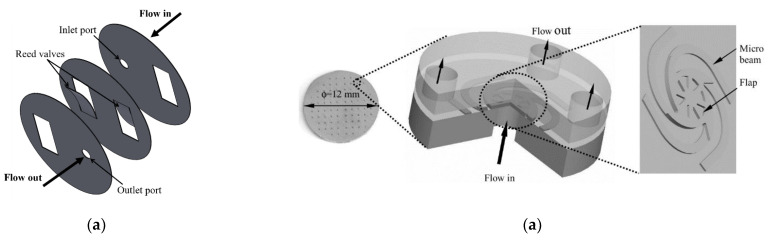
(**a**) Structure of cantilever reed valves. (**b**) Structure of micro arm valves (from [177]).

**Figure 16 micromachines-12-00140-f016:**
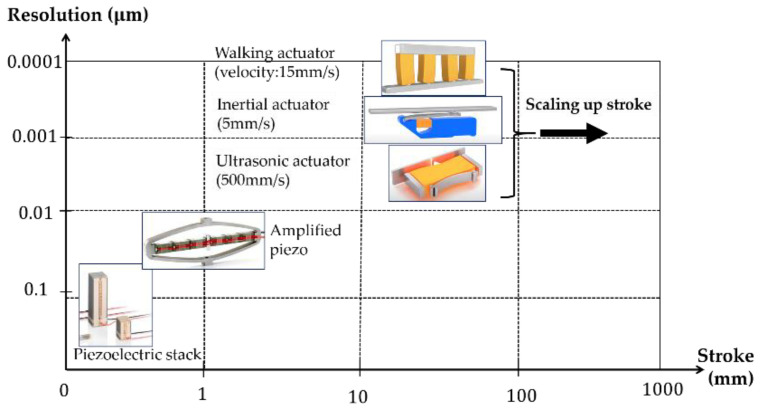
Summary of the strokes and resolution of some commercial piezoelectric-based products.

**Figure 17 micromachines-12-00140-f017:**
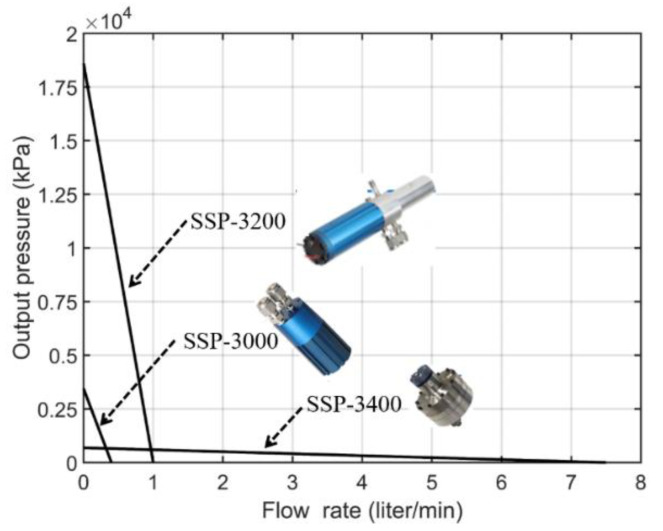
Summary of the performance of commercial piezoelectric-hydraulic pumps from Kinetic Ceramics company (solid-state pump SSP) (from [164]).

**Figure 18 micromachines-12-00140-f018:**
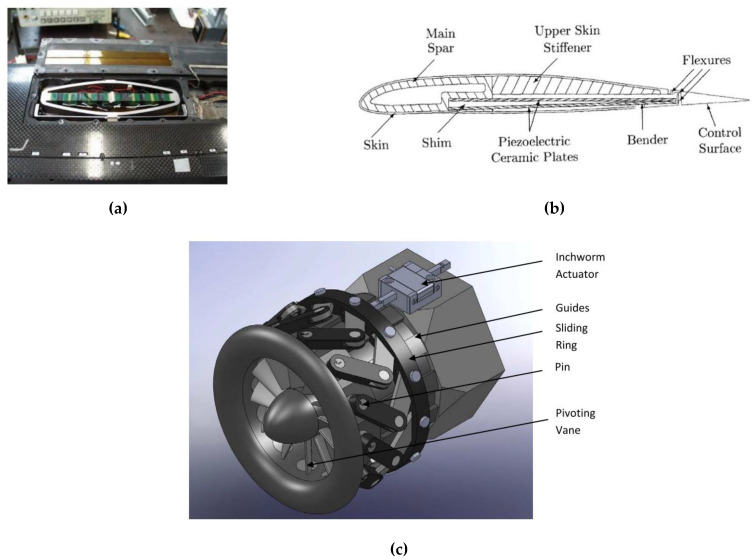
Examples of the piezoelectric system in aerospace applications: (**a**) active flap on a helicopter blade using an amplified-piezoelectric actuator [183]; (**b**) piezoelectric bender for helicopter rotor control (from [100]); (**c**) linear inchworm piezoelectric actuator for application in inlet guide vanes (from [47]).

**Table 1 micromachines-12-00140-t001:** Examples of some properties of piezoelectric materials.

Materials	$d_{33}$ at Room Temp. (pC/N)	Curie Temp. *T*_C_ (°C)	Operating Temp. (°C)	Reference
PZT powder	590–610	-	-	[20]
PMN-PT	2000–3500	120–130	Up to 80	[42,60]
PZN-PT	1900–2000	160	Up to 110	[42,64]
PZT-5H	585	170	−150–125	[44]
PZT Navy Type III (Hard) ^1^	<300	305	Up to 220	[45]
PZT-4	225	310	−150–100	[44]
PZT Navy Type II (Soft) ^2^	<600	340	Up to 200	[45]
PZT-5A	350	350	Up to 250	[44]
PIC series	240–500	160–370	−40–150	[65]
Lead-free materials				
BTBK	58.9–117	170–223	-	[56]
BNT	91	320	-	[56]
KNN	80–160	Up to 400	-	[57]
High-temperature materials				
Pb(NbO_3_)_2_	81	550	Up to 300	[45]
Bi_4_Ti_3_O_12_	3.5	675	Up to 675	[45]
Bi_4_Ti_2.86_Nb_0.14_O_12_	20	655	Up to 655	[45]

^1^ Hard: less hysteresis loss, good stability under high mechanical loads, and operating field strength; thus good for ultrasonic transducers. ^2^ Soft: large hysteresis loss, large piezoelectric charge coefficient, and easy polarization at low field strength; thus ideal for actuator and sensor applications.

**Table 2 micromachines-12-00140-t002:** Typical displacement and resonant frequency of typical geometrical forms of commercial piezoelectric actuators.

Form	Typical Size	Displacement Range	Resonant Frequency
Single layer (wafer)	A few hundred micrometer thickness	Up to 0.1 μm	Up to 100 kHz
Multilayer extension (rectangle, round, hollow stacks)	$\mathrm{Up}\;\mathrm{to}\;100\;{\mathrm{mm}}^{2}$ area & 100 mm in length	Up to 100 μm	Up to 100 kHz and more
Multilayer shear ing	$\mathrm{Up}\;\mathrm{to}\;\sim 250\;{\mathrm{mm}}^{2}$area & 50 mm length	Up to 10 μm	Up to 100 kHz and more
Piezo bender (unimorph/bimorph)	<1 mm thickness	10 μm to 2 mm	Up to a few kilohertz
Macrofiber composite (elongation)	Up to 140 mm active length	Up to 150 μm ^1^	From kIlohertz to megahertz
Macrofiber composite (contraction)	Up to 170 mm active length	Up to 100 μm ^2^	From kIlohertz to megahertz

^1^ Free strain: up to 1050 ppm. ^2^ Free strain: up to −600 ppm.

**Table 3 micromachines-12-00140-t003:** Typical input signals and related power consumption for piezoelectric actuators.

Sinusoidal Waveform	Square/Rectangle Waveform	Sawtooth Waveform
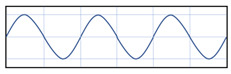	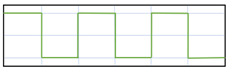	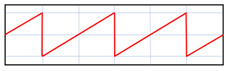
$P=\pi fCV_{pp}{}^{2}$/4	$P=fCV_{pp}{}^{2}$	$P=fCV_{pp}{}^{2}/\sqrt{3}$

where f is the working frequency, C is the capacitance of the piezoelectric, and $V_{pp}$ is the applied peak to peak voltage. The piezoelectric heat dissipation is usually 10% of the power supplied to the load. Therefore, the selection of usage piezoelectric materials and operating conditions must be weighed against the consumed power to ensure that the system’s power budget is optimized.

**Table 4 micromachines-12-00140-t004:** Performance of a typical piezoelectric-based actuator.

Flexure Hinge	Piezoelectric (mm)	Voltage (*V*p-p)	Frequency (Hz)	Force (N)	Speed (mm/s)	Reference
Symmetrical (Z shaped)	5×5×20 (Two)	100	5000	3.43 ^1^ (1 Hz)	6.057	[105]
Symmetrical (bridge type)	5×5×20 (Two)	100	1000	1.58 (1 Hz)	7.95	[104]
Asymmetrical (nonparallel type)	5×5×20 (One)	100	500	2.94 *	5.96	[92]
Asymmetrical (parallelogram type)	5×5×20 (One)	100	2000	3.43	14.25	[107]
Asymmetrical (four-bar mechanism)	5×5×20 (One)	100	490	4.32	15.04	[93]

^1^ Force (N) = weight (kg) × 9.81.

**Table 5 micromachines-12-00140-t005:** Comparison of the output force and speed performance of three commercial designs using the walking mechanism.

Configuration of the Piezoelectric Stack	Output Force	Speed	Resolution
Bending configuration (NEXACT)	~10–20 N/50 g (20 mm travel range)	Medium Up to 10 mm/s	0.03 nm (open loop) -
V-shape configuration (PICMAWalk)	~50–60 N/700 g (20 mm travel range)	High Up to 15 mm/s	0.02 nm (open loop) <10 nm (closed loop)
Combination of longitudinal and shear modes (NEXLINE)	Up to 600 N/1250 g (20 mm travel range)	Low Up to 1 mm/s	0.03 nm (open loop) 5 nm (closed loop)

**Table 6 micromachines-12-00140-t006:** Properties of typical working fluids.

Fluid	Density (kg/m^3^)	Temperature Range (°C)	Bulk Modulus (GPa)	Viscosity		Reference
				40 °C	100 °C	
Water	997	0–100	2.1	0.7	0.5	[171]
70% Glycerinaquerous	1181	−39–114 ^1^	0.4	22.5	-	[172]
Hydrolubric 123-B	-	~1 (Pour point)	-	21.5 ^3^	-	[173]
Mobile DTE-24	871	−27–220 ^1^	1.7	31.5	5.3	[174]
AeroShell 41	874	−41–135 ^2^	-	15.68	6.13	[175]
MIL-H5606F	859	−54–135	1.79	15	-	[154]
IL-EMIM-EtSO_4_	1241	162 (Flash point)	3.1	39.44	7.66	[170]

^1^ From pour point to flash point. ^2^ At the pressurized condition. ^3^ At 100 F (~38 °C).

**Table 7 micromachines-12-00140-t007:** Comparison of valve types in the piezoelectric–hydraulic pump.

Type	Pros.	Cons.
Reed valve	Simpler structure	Working frequency limitations
Microreed valve array	Small size and low inertia, hence a broader working frequency range	Complex structure and requirement of micromachining
Active valve (piezoelectric disc)	Operates at a higher frequency	Complex control
Diffuser valve—conical shape	No moving part, hence no fatigue failure	Leakage
Diffuser valve—Tesla valve	Lesser pressure drop	Complex structure

**Table 8 micromachines-12-00140-t008:** Summary of the performance of several piezoelectric–hydraulic pumps.

Piezo Stack (mm)	Voltage (*V*p-p)	Frequency (Hz)	Valve	Pressure (kPa)	Flow Rate (mL/min)	Reference
PZT stack 19 × 19 × 102	800	60	Commercial ball-type check valve	3800	312	[141]
PZWT100 ∅13 × 20	1000	1000	Unimorph disc valve	8300	204	[165]
P-885.91 7 × 7 × 36	120	60	Reed valve	7.96	10.32	[169]
APC Pst150 10 × 10 × 81	100	200	Reed valve	550	1140	[139]
P-804.10 (×2) 10 × 10 × 18	100	300	Reed valve	1600	180	[154]
APC Pst150 3.5 × 3.5 × 18	150	400	Reed valve	125	186	[167]
EPCOS PZT (×3) 6.7 × 6.7 × 30	150	400	Reed valve	1724	338	[182]
P-025.40P ∅25 × 60	1000	300	Reed valve (double piezo pump)	6532	1246	[163]
PZWT100 ∅81 (120 μm stroke)	1200	1000	Reed valve MEMS valve	6895	1830	[168]

**Table 9 micromachines-12-00140-t009:** Comparison of piezoelectric-based systems.

Concept	Structure	Force	Stroke	Resolution	Speed	Control
Continuous motion						
Flexure mechanism	Medium	Medium	Small	-	-	Simple
Lever, X-mechanism	Simple	Medium	Small	-	-	Simple
Bimorph configuration	Simple	Low	Very small	-	-	Simple
Quasi-static stepped motion						
Inchworm—intermittent	Medium	Large	Large	Medium	Low	Complex
Inchworm—continuous	Medium	Medium	Large	Medium	Low	Complex
Inertial	Medium	Small	Large	High	Medium	Simple
Walking—bending legs	Complex	Small	Large	High	High	Medium
Walking—V-shape legs	Complex	Small	Large	High	High	Medium
Walking—combination legs	Complex	Medium	Large	High	Medium	Medium
Ultrasonic stepped motion						
Standing wave	Medium	Small	Large	Medium	High	Medium
Traveling wave	Complex	Small	Large	Medium	High	Complex

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
