# Peer review of "Large-Scale Piezoelectric-Based Systems for More Electric Aircraft Applications"

_micromachines, 2021, doi:10.3390/mi12020140_

Round 1
Reviewer 1 Report
Line 25-26 It is stated that "A concept of “More/All Electric Aircraft” has recently received huge attention in Research and Development work in the field of Aerospace Engineering [1-3]" but the publications published in 1984, 2007, 2017. The latest published in SPIE proceeding and do not think that is enough to declare actuality of review paper.
There are just a few papers cited related to aircraft engineering by the usuage of piezo material.
Nowadays we have a lot of novel piezo composite materials with piezo effect.
Maybe it would be useful to make some overview of such kind of publication.
Maybe paper would be more richer if would be presented how displacement and vibration amplitude depends on geometrical form of piezo element.I didn't find any consideration about accuracy of piezodevices.
I reference Nr. 7 year of paper publishing is missing.
Author Response
Dear Reviewer,
Thank you for your kind comment that helped us.
I've attached the point by point response to your comment (see attached). It also includes the comment by the other reviewers.
Thank you for your support
Holden Li

Reviewer 2 Report
The contribution "Large-scale Piezoelectric-based systems for “More Electric Aircraft” applications" is a very good motivating review with pedagogical structure, e.g. theory, state-of-the-art, commercial applications. However, the review somehow gives the impression that wants to find a piezo-based solution that fits all the applications for electric aircraft, which unfortunately is difficult. Thus, it would have been much better to have also the aircraft applications' requirements / specifications as starting point for this review. It is also difficult to see from the contribution i) for which aircraft applications is the piezoelectricity an advantage and ii) which are the considerations in their review for 'novel' designs in aerospace applications. The authors themselves give the essential disadvantages for using piezo-based components for such critical application where (small) failures are not allowed.
For a proper review article having the current title, it should contain some starting considerations on requirements / specifications specifically for aircraft applications from where the review should begin looking to various piezo-based design solutions. Moreover, the descriptions are quite qualitative making difficult to get suitable solutions.
Author Response

(The authors gave the same response as above.)

Reviewer 3 Report
- The unit of temperature in the manuscript is error, please change its format;
- Some new research references should be cited.
Author Response

(The authors gave the same response as above.)

Round 2
Reviewer 1 Report
The authors of the paper did revision of the paper according recommendations.
In my opinion paper may be published.
Reviewer 2 Report
The authors have improved very well the contribution according to the suggestions.